Parametric estimation of P(X > Y) for normal distributions in the context of probabilistic environmental risk assessment

Jacobs Rianne 1 rianne.jacobs@wur.nl
Bekker Andriëtte A. 2
van der Voet Hilko 1
ter Braak Cajo J.F. 1
1 Biometris, Wageningen University and Research Centre , Wageningen , The Netherlands
2 Department of Statistics, University of Pretoria , Pretoria , South Africa
Minasny Budiman
Electronic publication date: 2015 Aug 18
Publication date: 2015
Volume: 3
Electronic Location ID: e1164
Received 2015 Jun 1; Accepted 2015 Jul 17
Copyright: © 2015 Jacobs et al.
Copyright year: 2015
Copyright holder: Jacobs et al.
License: This is an open access article distributed under the terms of the Creative Commons Attribution License, which permits unrestricted use, distribution, reproduction and adaptation in any medium and for any purpose provided that it is properly attributed. For attribution, the original author(s), title, publication source (PeerJ) and either DOI or URL of the article must be cited.
License URL: https://creativecommons.org/licenses/by/4.0/

Keywords: Bayesian estimator, Maximum likelihood estimator, Risk assessment, Simulation, Nanoparticle

Funding: NanoNextNL This work is supported by NanoNextNL, a micro and nanotechnology consortium of the Government of the Netherlands and 130 partners. The funders had no role in study design, data collection and analysis, decision to publish, or preparation of the manuscript.

==============================
Estimating the risk, P(X > Y), in probabilistic environmental risk assessment of nanoparticles is a problem when confronted by potentially small risks and small sample sizes of the exposure concentration X and/or the effect concentration Y. This is illustrated in the motivating case study of aquatic risk assessment of nano-Ag. A non-parametric estimator based on data alone is not sufficient as it is limited by sample size. In this paper, we investigate the maximum gain possible when making strong parametric assumptions as opposed to making no parametric assumptions at all. We compare maximum likelihood and Bayesian estimators with the non-parametric estimator and study the influence of sample size and risk on the (interval) estimators via simulation. We found that the parametric estimators enable us to estimate and bound the risk for smaller sample sizes and small risks. Also, the Bayesian estimator outperforms the maximum likelihood estimators in terms of coverage and interval lengths and is, therefore, preferred in our motivating case study.

Introduction

Like all novel materials, engineered nanoparticle (ENPs) have no history of safe use. A risk assessment is important for the societal acceptance and safe use of ENPs. In order to perform a proper risk assessment, one needs knowledge and data on the properties of nanoparticles. These properties can be different in nanoparticles compared to conventional chemicals in areas such as physicochemical properties, life cycle, toxicokinetics and environmental fate. This information is hard to come by because of lack of knowledge and technical limitations, resulting in no or only small datasets for effect concentrations of ENPs. In the EU, environmental risk assessment is regulated by the European Chemicals Agency (ECHA) and probabilistic risk assessment is level 3 of their tiered risk assessment approach (ECHA, 2012).

In the motivating case study of aquatic risk assessment of nano-Ag (Gottschalk, Kost & Nowack, 2013), we are confronted with such a small dataset of effect concentrations. Gottschalk, Kost & Nowack (2013) modeled the exposure of nano-Ag from surface water with a probabilistic material-flow model (Gottschalk, Scholz & Nowack, 2010) to obtain a distribution of exposure concentration values. They collected the effect concentration data from available toxicity studies found in the literature. These effect concentration data consist of toxic endpoints (eq. LC50, EC50, NOEC) for 12 aquatic species. For some of these species there were more than one data point. We averaged these to obtain one value for each species. Histograms and normal density curves of the exposure and effect concentration data are given in Fig. 1.

Figure 1 Histograms and normal density curves of exposure (nx = 1,000) and effect (ny = 12) concentration nano-Ag (µg/L).

Data taken from Gottschalk, Kost & Nowack (2013).

In probabilistic risk assessment, the variability of environmental exposure due to natural variation in concentration values over various environments is modelled by an exposure concentration (X) distribution (ECD). Similarly, the variability in effect concentration values due to natural variation among species in their sensitivity to nanoparticles is modelled by a species sensitivity distribution (SSD) or effect concentration (Y) distribution. Probabilistic risk estimation is based on the overlapping of the ECD and the SSD (ECHA, 2012). The risk, R = P(X > Y), is the area under the curve obtained by multiplying the probability density function (pdf) of the ECD with the cumulative distribution function (cdf) of the SSD. Verdonck et al. (2003) critically discuss this approach to risk assessment.

In the ecotoxicological risk assessment literature, R = P(X > Y) as a definition for risk was first developed by Suter, Vaughan & Gardner (1983). This concept was further developed by Van Straalen (2002) as ecological risk (ER). A similar concept, known as expected total risk, was developed by the Water Environment Research Foundation (WERF) (Cardwell et al., 1993; Warren-Hicks, Parkhurst & Butcher, 2002). For a visual representation of risk, the exceedance function (1-cdf) of the ECD is plotted against the cdf of the SSD. Such a plot is called a joint probability curve (JPC) (ECOFRAM, 1999; Solomon, Giesy & Jones, 2000; Solomon & Takacs, 2002). The area under the JPC is the risk (Solomon & Takacs, 2002). Aldenberg, Jaworska & Traas (2002) showed that the area under the JPC, ER and expected total risk are mathematically identical. In this paper, we refer to the probability, P(X > Y), as the risk, R.

When we consider the case study data, there is no overlap between the effect and the exposure histograms (Fig. 1). There is no exposure concentration that is greater than an effect concentration, and, therefore, the empirical estimate of R = P(X > Y) is zero for these datasets. To conclude, however, that the true R is zero based on a small sample is rather imprudent. The denial of a probability of zero is referred to as Cromwell’s rule by Lindley (1971, p. 105–106; 2006, p. 90–91). Several possibilities exist to address the zero problem empirically. This is further discussed in Section ‘Non-parametric estimator’. The zero problem can also be solved by fitting a parametric distribution to the data. When considering the normal density curves in Fig. 1, we note that there is some overlap between the exposure and effect concentration distributions and, therefore, some non-zero probability of exposure values exceeding effect values.

It is common to assume independent log-normal distributions for the exposure distribution and the species sensitivity distribution (SSD) (Aldenberg, Jaworska & Traas, 2002; Verdonck et al., 2003; Wagner & Løkke, 1991). This is the same as assuming normal distributions on the log-transformed exposure and effect concentrations. This normal–normal model was developed in some detail by Aldenberg, Jaworska & Traas (2002) and allows an analytic expression for the risk when parameter values are known.

Estimation of R = P(X > Y) is also of interest in other areas such as engineering and medical applications. In engineering, R = P(X > Y) is known as the reliability in stress–strength models. This is a well-known concept and has been studied extensively for the normal–normal model (Barbiero, 2011; Church & Harris, 1970; Downtown, 1973; Enis & Geisser, 1971; Govidarajulu, 1967; Nandi & Aich, 1996; Voinov, 1986; Weerahandi & Johnson, 1992) as well as for other distributions (Kundu & Gupta, 2006; Mokhlis, 2005; Nadar, Kızılaslan & Papadopoulos, 2014). None of these papers give sufficiently general theory for obtaining trustworthy interval estimates in the case study. In receiver operating characteristic (ROC) analysis such as used in medical applications, P(X > Y) is known as the area under the ROC curve (Li & Ma, 2011). Although usually used for categorical data, the area under the ROC curve can also be obtained for continuous data in both a non-parametric way and for the normal–normal model (Krzanowski & Hand, 2009).

In this paper, we will investigate the influence of sample size on the estimation of R = P(X > Y), with special attention to the sample size of effect concentrations. We also investigate the behaviour of the estimators of R = P(X > Y) for small risks. We consider one non-parametric estimator and three parametric estimators, namely, the maximum likelihood estimator (MLE), quasi maximum likelihood estimator (QMLE) and Bayesian estimator with noninformative prior for the normal–normal model. In comparing the parametric estimators with the non-parametric one, we investigate the maximum gain possible when making strong parametric assumptions as opposed to making no parametric assumptions at all. This is done in a simulation study in which we also assess the accuracy and precision of the estimators and compare them for various combinations of sample sizes and risks.

In ‘Theory and Methods’, we derive the estimators and provide the simulation structure. In ‘Simulation Results’, the simulation results will be given and discussed. In ‘Case Study’, the application is discussed in the context of the simulation results. ‘Discussion and Conclusion’ provides some general discussion, conclusions and recommendations for further study.

Theory and Methods

In this section, we describe the theory and methodology of our approach. We start by deriving the risk for the normal–normal model. Next we discuss the four estimation methods, provide the simulation structure and discuss the performance measures that we used.

Risk

Let X be the log10 exposure concentration random variable and Y be the log10 effect (or log10 sensitivity) concentration random variable.

In the normal–normal model, the distributions are given by X∼Nμx,σxY∼Nμy,σy.

Due to the additive property of the normal distribution we have X−Y∼Nμx−μy,σx2+σy2.

The risk for the normal–normal model is given by (1) R=PX>Y=PX−Y>0=1−Φ−μx−μyσx2+σy2=Φμx−μyσx2+σy2

where Φ(⋅) denotes the standard normal distribution. Equation (1) is a well-known result (Reiser & Guttman, 1986).

We note location-scale invariance in Eq. (1). The value of R is determined only by the difference of the expected values and the sum of the variances. The absolute value of the individual parameters is not relevant.

Point estimation

In the following sections, we derive the MLE, QMLE, Bayesian estimator and non-parametric estimator for the risk, R. We let (x1, x2, …, xnx) be a random sample of size nx of log exposure concentrations and (y1, y2, …, yny) an independent random sample of size ny of log effect concentrations.

Maximum likelihood estimator

The most straightforward way of estimating R is by means of maximum likelihood estimation. The estimator obtained in this way is denoted as RˆMLE. From the invariance property of MLEs (Bain & Engelhardt, 1992, p. 296), we obtain RˆMLE by substituting the MLEs of μx, μy, σx2 and σy2 in Eq. (1). These MLEs are given by

Parameter	Maximum likelihood estimator	
μx	x¯=1nx∑i=1nxxi	
μy	y¯=1ny∑i=1nyyi	
σx2	σˆx2=1nx∑i=1nxxi−x¯2	
σy2	σˆy2=1ny∑i=1nyyi−y¯2	

Equation (1) then becomes (2) RˆMLE=Φx¯−y¯σˆx2+σˆy2.

Note that σˆx2 and σˆy2 are the MLEs of the variance, which are not unbiased.

Quasi maximum likelihood estimator

The QMLE is similar to the MLE, differing only in the use of unbiased estimators for σx2 and σy2 instead of the MLEs. We then obtain (3) RˆQMLE=Φx¯−y¯sx2+sy2

with sx2=1nx−1∑i=1nxxi−x¯2 and sy2=1ny−1∑i=1nyyi−y¯2.

Bayesian estimator

Our third way of estimating R is Bayesian. Whereas maximum likelihood estimation uses the data only, Bayesian estimation combines prior knowledge about the parameter(s) with the data. The prior knowledge is specified by a prior distribution and the information in the data by the likelihood. The prior distribution and the likelihood are then combined into what is called the posterior distribution of the parameter (Gelman et al., 2014). We will derive the joint posterior distribution of the parameters μx, μy, σx2 and σy2. This distribution together with Eq. (1) will provide us with the posterior distribution, fR(r), of R.

Unfortunately we have often very little prior knowledge. Therefore, we derive RˆBayes assuming a non-informative prior distribution for the parameters, namely pμx,σx2∝1σx2 for the joint prior distribution of μx,σx2 and pμy,σy2∝1σy2 for μy,σy2 (Gelman et al., 2014, p. 64).

The posterior distributions are then given by (Gelman et al., 2014, p. 65)

• μx∣σx2∼Nx¯,σxnx

• μy∣σy2∼Ny¯,σyny

• σx2∼Inverse-gammanx−12,nx−1sx22

• σy2∼Inverse-gammany−12,ny−1sy22.

From this, we obtain the conditional posterior distribution of μx−μyσx2+σy2 (Weerahandi & Johnson, 1992) μx−μy∣σx2,σy2∼Nx¯−y¯,σx2nx+σy2ny

∴μx−μyσx2+σy2∣σx2,σy2∼Nx¯−y¯σx2+σy2,σx2nx+σy2nyσx2+σy2.

Using the variable transformation method and integrating σx2 and σy2 out of the joint pdf, fr,σx2,σy2, we obtain the marginal pdf, fR(r) (see Appendix Result A.2 for details). This marginal posterior pdf of R (Eq. (A.4)) can be evaluated using numerical integration.

Alternatively, we can use Monte Carlo sampling to approximate the marginal posterior pdf of R. We used the Method of Composition (Lesaffre & Lawson, 2012, p. 93–94) in which we sample from the known posterior distributions of σx2 and σy2, then μx and μy from their known posterior conditional distributions and then apply Eq. (1) to obtain the corresponding R value. Figure 2 shows histograms of samples drawn from the marginal posterior distribution of R (sample size of 10,000) together with the marginal pdf computed by numerical integration using Eq. (A.4) for different sample sizes and R values. It can be seen that the Monte Carlo method gives a very good approximation to the theoretical posterior pdf evaluated by numerical integration.

Figure 2 Histogram and theoretical posterior pdf (black solid line) of R.

Sample sizes of 5 and 20 for effect concentrations, sample sizes of 5 and 100 for exposure concentrations and R value of 0.1 and 0.5.

In this paper, we obtained the posterior distribution of R by sampling, because it required less computing time than numerical integration in our implementation. The posterior mean is often taken as the Bayesian point estimator but we also investigated the posterior median and mode as point estimators of R.

Non-parametric estimator

As a benchmark comparison for the parametric estimators, we included a basic non-parametric estimator. This estimator, RˆNP, is calculated from the data without any distributional assumptions by (4) RˆNP0=1nxny∑i=1nx∑j=1nyIxi>yj+12Ixi=yj

where IS=1 if S is true and 0 otherwise (Krzanowski & Hand, 2009, p. 65). Alternatively, Eq. (4) can be written as RˆNP0=Unxny

where U is the Mann–Whitney statistic (Gibbons & Chakraborti, 2011). Equation (4) is also known as the area under the ROC curve and its equivalence to the Mann–Whitney statistic has been shown (Bamber, 1975).

The non-parametric estimator of R is related to estimating the success probability, p, in a binomial experiment. As noted in the Introduction, we encounter the zero problem. One possible solution is making use of Laplace’s Law of Succession (Zabell, 1989). This law states that given k successes in n trials of a binomial experiment, the probability of a success on the next trial is k+1n+2. The validity of this expression has a Bayesian basis. On assuming a uniform prior for p, the posterior distribution of p is a Beta(k + 1, n − k + 1) distribution (Gelman et al., 2014, p. 30), so that the posterior mean is k+1n+2. This expression, denoted RˆNPLLS, is then used instead of the estimator in Eq. (4). Note that the posterior mean is equal to the predictive probability of a success on the next trial.

An alternative solution is to replace the zero with some non-zero value. One option is to estimate the probability of an outcome outside the range of the data as 12nxny. This method is used by Matlab and Genstat to compute quantiles. Another alternative is to use 1nxny+1 which is used by Minitab and SPSS.

Interval estimation

We propose interval estimators by calculating credible intervals for Bayesian methods and confidence intervals for others. For the Bayesian estimator, we calculated 90% two-sided highest posterior density (HPD) credible intervals (Box & Tiao, 1973, p. 123). These intervals are obtained by finding the interval of the posterior distribution with the highest density, for which we used the ‘HPDinterval’ function in the ‘coda’ package in R (Plummer et al., 2006). HPD intervals produce the shortest intervals on a chosen scale, e.g., R or a transformation thereof, but are not transformation invariant. To estimate an upper credible bound of the risk, we also calculated the 95% percentile of the posterior. The upper bound of the 90% two-sided HPD intervals is not necessarily equal to the 95% percentile as the probabilities to the left and right of the two-sided HPD interval can be unequal.

For the non-Bayesian estimators, we calculated 90% Bias corrected and accelerated (BCa) parametric bootstrap confidence intervals using the ‘boot’ package in R (Canty & Ripley, 2013; Davison & Hinkley, 1997) with 1,000 bootstrap samples. For the non-parametric estimator, the BCa interval algorithm did not converge for small R values and also had some difficulty with the small sample sizes. For these cases we calculated percentile confidence intervals. The percentile method obtains a symmetric 100(1 − α)% confidence interval by calculating the α2th and 1−α2th percentiles of the bootstrap sample. In the BCa method, these percentiles are adjusted to correct for bias and skewness. For symmetric distributions, the percentile and BCa intervals are equal. Both intervals are also transformation invariant (Efron & Tibshirani, 1993, p. 175, 187). When calculating the confidence intervals for small risks, all bootstrap values may be equal, resulting in a zero interval length. For the BCa and percentile interval, the upper bound of a 90% two-sided interval is equal to the upper bound of a 95% one-sided interval (Carpenter & Bithell, 2000). The upper 95% confidence bound is, therefore, trivially obtained from the 90% two-sided interval.

For the MLE-like estimators, we also calculated confidence intervals based on the noncentral t distribution (Reiser & Guttman, 1986). In this method, the sum of the two variances (sx2 and sy2) are approximated with a chi-squared distribution.

The upper confidence (credible) bounds are of special interest in the context of managing risks, as they indicate (with some certainty) that the risk will not be higher than the upper bound.

Simulation study

In this section, we discuss the design of the Monte Carlo (MC) simulation study following the guidelines provided in Burton et al. (2006).

Simulation setup

The simulation study is performed in R (R Core Team, 2013). We use the built-in rnorm function to sample from a normal distribution using the Mersenne-Twister pseudo-random number generator (Matsumoto & Nishimura, 1998). Starting seeds for the different scenarios were drawn from a discrete uniform distribution to produce independent samples for each sample size scenario. The four estimators are calculated on the same sample, thereby avoiding differences among the estimators due to sampling.

Table 1 Substances, sample sizes and estimated risks in an environmental risk assessment performed by Gottschalk, Kost & Nowack (2013).

Sample size of effect concentration data are given for aquatic and soil toxicity. Risks are given for four environmental compartments.

Substance	Sample size	Risks	
	Aquatic	Soil	1	2	3	4	
Ag	12	1	0.007	0.397	0	0	
CNT	9	2	0	0	0	0	
TiO2	18	2	7.2e−13	0.187	0	1.2e−7	
ZnO	17	2	0	0.011	0	0	
Fullerenes	4		0	0	0	0	

To make the simulation as realistic as possible, we chose scenarios that are in line with recent studies of environmental risk assessment. When exposures are measured, it is common to have small sample sizes (Johnson et al., 2011; Westerhoff et al., 2011), whereas any number of exposure values can be obtained when they are modeled (Gottschalk, Kost & Nowack, 2013). For the exposure sample size, therefore, we chose two scenarios: the case of a small number of exposures (nx = 5) and the case of a (relatively) large number of exposures (nx = 100). We chose sample size of effect concentrations and risks loosely suggested by data from Gottschalk, Kost & Nowack (2013). From this data (Table 1), we chose the following scenarios:

• Sample sizes for effect concentrations (ny): 2, 5, 12, 20, 100

• risks: a grid of values from 1e−14 to 0.5.

The sample size of 100 for effect concentrations was added to study the influence of a large sample size. A risk of 0.5 is obtained when μx = μy. We, therefore, chose increasing values of μx − μy to obtain the required range of risks. Considering the standard deviations, we note that the standard deviation of the effect concentration data in the case study is 5.6 times larger than that of the exposure concentration data. Based on this, we chose three scenarios: σy = σx, σy=15σx and σy = 5σx.

The number of simulations was determined by running a pilot simulation (1,000 simulations) for the MLE. From this pilot, we obtained the median empirical standard deviation (sd = 0.0719496) of RˆMLE and the median absolute bias (δ = 0.002107476) in RˆMLE over all scenarios. These were used to calculate the number of simulations, B, according to Burton et al. (2006) B=z0.95sdδ2=1.96⋅0.07194960.0021074762=4477.58≈4,500.

For each of the 4,500 MC simulations, we calculated RˆMLE, RˆQMLE, RˆBayes and RˆNP. Due to the skewness of their sampling distributions, especially for small R values, we decided to use a transformation. Due to the nature of the analytical expression for R, (see Eq. (1)), a probit (inverse standard normal cdf) transformation is a natural choice. Some comparisons between the original scale and the probit transformation are further discussed for the Bayesian case in Section ‘Comparison of Bayesian point estimators’.

Simulations were run on a HP desktop computer running Microsoft Windows 7 with processor specification Intel(R) Core(TM) i7-4770 CPU @ 3.40GHz, 3401 MHz, 4 Core(s), 8 Logical Processor(s). Three R-sessions were running the three cases σy = σx, σy=15σx and σy = 5σx simultaneously. The σy = σx case took the longest with the following time (in hh: mm:sec) for each of the four estimators:

• MLE: 03:43:06.18 (bootstrap); 00:10:21.82 (noncentral t)

• QMLE: 03:35:36.44 (bootstrap); 00:10:13.30 (noncentral t)

• Bayes: 03:12:56 (sample size 10,000); 00:41:02.9 (sample size 1,000)

• Non-parametric: 19:13:17.12

The bootstrap of the MLE, QMLE and non-parametric estimator was the cause of the longer runtime. The runtime for the Bayesian estimator is directly related to the size of the posterior sample. All further results are given for the large sample case.

Performance measures

We calculated various performance measures to evaluate the performance of the four point estimators. We calculated the performance measures on the probit scale, so as to be able to highlight differences among methods for small values of R:

• mean: probitRˆ¯=∑i=14,500probitRˆi4,500

• absolute bias: bias=probitRˆ¯−probitR

• empirical (or MC) standard deviation: SD=∑i=14,500probitRˆi−probitRˆ¯24,500

• root mean squared error: RMSE=bias2+SD2.

The quality of the interval estimators will be assessed by calculating the coverage probability for each scenario. Note that the confidence intervals we calculated are approximate and do not claim to deliver the correct coverage. In addition, Bayesian credible intervals also do not claim a coverage frequency. For each of the 4,500 simulations, we calculated the confidence (credible) intervals and calculated the proportion of intervals that contained the true R value. We also investigated lengths of confidence (credible) interval over the 4,500 simulations for each scenario and R value. Coverages and lengths of confidence intervals for the different non-parametric estimators were similar. We, therefore, only consider the estimator based on Laplace’s Law of Succession.

Simulation Results

All results given and discussed are for the scenario that most resembles the case study (σy = 5σx and sample sizes nx = 100 and ny = 12) unless explicitly stated otherwise. Graphs and tables for the other scenarios are given in the Supplemental Information. All sampling distribution graphs plot the estimated probitRˆ (or Rˆ) value (y-axis) against the true probit(R) (or R) value (x-axis). A diagonal 1–1 line is drawn to indicate where probitRˆ=probitR (or Rˆ=R). A logarithmic scale is used when R is plotted.

Comparison of Bayesian point estimators

For Bayesian estimation, we considered three point estimators, namely, the posterior mean, median and mode. We summarize the sampling distribution of each by way of three quantiles (0.5 or median, 0.025 and 0.975). In Fig. 3 these quantiles are plotted as a function of the true value. Figures 3A and 3C show the median and Fig. 3B and 3D show the 0.025 and 0.975 quantiles.

Figure 3 Quantiles of the sampling distribution of the three Bayesian point estimators (mean, median and mode).

The median (A, C) and 0.025 and 0.975 quantiles (B, D) of the sampling distribution of the three Bayesian point estimators (mean, median and mode) calculated on the original scale (A, B) and on the probit scale (C, D). When plotted on log10-scale, a zero mode becomes −∞. The diagonal dotted line represents the values where probitRˆ=probitR (or Rˆ=R).

In Figs. 3A and 3B, the quantiles are calculated from the sampling distribution of the estimators on the original scale (Rˆ) and plotted on log10-scale. The lines for the posterior mean are above the 1:1 line, so indicating large positive bias. The lines for the posterior mode go to log10(0) = − ∞, due to the very skew posterior distributions for smaller risks (on original scale of R, as already illustrated in Fig. 2), so indicating large negative bias. The lines for the posterior median are in between and closer to the 1:1 line.

In Fig. 3C and 3D, the quantiles are calculated similarly but on the probit-transformed Rˆ and plotted on the probit scale as well. Here we see that quantiles of the posterior mean, median and mode almost coincide, indicating that the skewness problem is solved. Simulations for very small sample size of effect concentrations (ny = 2) showed that the mean has a slight advantage because of narrower intervals between the 0.025 and 0.975 quantiles than that of the median and the mode. This difference, however, is very quickly lost for higher sample sizes (ny ⩾ 5) as shown in Fig. S23.

From this study of Bayesian point estimators on different scales, we see the advantage of the use of the probit scale for the Bayesian case. Moreover, for ease and its transformation invariance, we chose the posterior median as the Bayesian point estimator of R. The probit scale stretches out small values of R, making possible differences between methods more clearly visible for small R. On this basis, we decided to perform, for all estimators, all further calculations in the simulation study on the probit scale.

Comparison of the four point and interval estimators

In this section, we show the simulation results for the four estimators.

We first compare the sampling distributions of the four estimators. Figure 4 illustrates the median (A) and 0.025 and 0.975 quantiles (B) of the sampling distributions of RˆMLE, RˆQMLE, RˆBayes and RˆNP. For RˆNP, we plotted both the standard estimator, RˆNP0, (Eq. (4)) which goes to minus infinity on the probit scale and the Laplace version, RˆNPLLS. These provide the extreme endpoints of the different solutions in solving the zero problem in the non-parametric estimator.

The median of the Bayesian estimator lies closest to the true R (Fig. 4A). This is especially apparent in scenarios with ny ≤ 12 (Fig. S24). The non-parametric estimators are clearly not able to estimate R for smaller values as they very quickly jump to their lower bound of either probit1nxxny+2 or minus infinity, indicated by horizontal and vertical dash–dot lines respectively. As the sample sizes increase, the three parametric estimators converge (Figs. S24 and S25). The non-parametric estimators remain the worst estimators for all sample sizes when estimating small R values. In our further study, we consider the Laplace version only.

Figure 4 Quantiles of the sampling distribution of the point estimators.

The median (A) and 0.025 and 0.975 quantiles (B) of the sampling distribution of the point estimators, RˆMLE, RˆQMLE, RˆBayes, RˆNPLLS and RˆNP0 calculated on the probit scale. The diagonal dotted line represents the values where probitRˆ=probitR.

Next, we study the coverage and interval lengths of the two-sided 90% confidence (credible) intervals of the estimators on the probit scale. For each of the 4,500 simulations, we calculated interval lengths and then obtained the median interval length for each combination of estimator, sample sizes and risk value combinations. In order to compare the median interval lengths across different risk values in a single graph, we standardized each one by dividing by the true probitR value to obtain the relative median interval length. Figure 5 plots the relative median interval length (y-axis) against the coverage probabilities (x-axis) for nx = 5 and nx = 100, all investigated sample sizes for effect concentrations, and all R values. Coverage probabilities of less than 0.5 were plotted at 0.5. The vertical line indicates a coverage probability of 90%. A good interval estimator gives points lying on this line with short interval length. This translates to good coverage and narrow intervals. Points corresponding to ny = 12 are indicated by an open black circle.

Figure 5 Scatterplots of the 90% two-sided coverage probabilities against the relative median interval length calculated on the probit scale.

The value of the true R value is illustrated by the color scale. The size of the dots corresponds to the size of the sample size of effect concentrations. A vertical reference line is drawn at a coverage probability of 90%. The points corresponding to ny = 12 are indicated by an open black circle.

We found that the MLE (not shown) and the QMLE had a similar pattern for both the bootstrap and noncentral t intervals, with the QMLE consistently having better coverage. Figure 5 shows that the bootstrap intervals have liberal coverage compared to the noncentral t intervals for small sample sizes of effect concentrations. As the sample sizes for effect concentrations increase (bigger dots), the estimators have better coverage. Very small sample size for effect concentrations (ny = 2) gives the worst coverage. For the Bayesian estimator, the sample size has a lesser influence. For small exposure sample size (nx = 5), the coverage of the Bayesian interval estimator tends to be too high. This problem largely disappears for nx = 100, although there is some under-coverage for the ny = 2 case. The parametric estimators have shorter interval lengths when nx = 100 (right column) as compared to nx = 5. For the non-parametric estimator, sample size has a slightly less systematic influence on the coverage.

Compared to the other estimators, the Bayesian interval estimator best maintains the nominal coverage without having larger median interval length and despite the fact of often having a higher than nominal coverage (Fig. 5). The non-Bayesian estimators have smaller than nominal coverage for small sample size of effect concentrations with the non-parametric estimator being the worst. For better comparison of interval lengths among the estimators, the reader is referred to Fig. S26.

In risk assessment, one is often interested in an upper bound on the risk. We studied the coverages and interval lengths of the upper 95% confidence (credible) bounds of the estimators on the probit scale. The interval lengths were quantified as the difference between the upper bound and the true probit(R) value. The median of the 4,500 differences was obtained. In order to compare the median differences across different risk values in a single graph, each median difference was divided by the true probitR value being estimated to obtain the relative median difference. Figure 6 plots the relative median difference (y-axis) against the coverage probabilities (x-axis) for nx = 5 and nx = 100, all investigated sample sizes for effect concentrations, and all R values. Coverage probabilities of less than 0.5 were plotted at 0.5. The vertical line specifies a coverage probability of 95%. A good upper bound estimator gives points lying on this vertical line and being close to the horizontal 0 line. This translates to good coverage and an upper bound close to the true R. Points corresponding to ny = 12 are indicated by an open black circle.

We see similar patterns as in the case of the two-sided intervals, with the bootstrap intervals being too liberal. The Bayes estimator gives higher than nominal coverage for small exposure sample size (left column). For large exposure sample size, the Bayesian estimator clearly outperforms the other estimators with good coverage for all R values and sample sizes for effect concentrations without having larger median interval difference. The non-parametric estimator has a severe coverage problem.

Figure 6 Scatterplots of the 95% one-sided coverage probabilities against the relative median difference calculated on the probit scale.

The value of the true R value is illustrated by the color scale. The size of the dots corresponds to the size of the sample size of effect. A vertical reference line is drawn at a coverage probability of 95%. The points corresponding to ny = 12 are indicated by a black circle.

The results for the performance measures of the different estimators are given in Tables S2–S5. Due to the lower bound of the non-parametric estimator, the SD, bias and RMSE are not reliable for small R values. Only for a few cases where R = 0.5, the non-parametric estimator has slightly lower SD and bias than the parametric estimators. The various graphs have also shown the inability of the non-parametric estimator to estimate small R values.

Among the parametric estimators, the Bayesian estimator as the smallest SD, bias and RMSE on probit scale for all sample sizes and R values. This confirms that the Bayesian estimator is better than the non-Bayesian estimators as also seen for the interval estimator case. The QMLE has smaller SD, bias and RMSE than the MLE for all sample sizes and R values. This also confirms the results of the interval estimators where QMLE has better coverage than MLE.

The Bayesian estimator was in general the best estimator and specifically so for the scenario that is closest to the case study. The Bayesian point estimator was less biased than the MLE and the QMLE in all cases (σy = σx (Figs. S8 and S9), σy=15σx (Figs. S16 and S17) and σy = 5σx (Figs. S24 and S25)), and this was especially apparent for small sample sizes for effect concentration (ny ≤ 12) and small R values. The Bayesian interval estimator (90% two-sided) had better coverage, with even higher than nominal coverage for exposure sample size nx = 5. This is also seen for the case σy = σx (Fig. S10). For the case σy=15σx (Fig. S18), the higher coverage is only seen for small sample size for effect concentrations as seen by the small dots. When considering the 95% upper bound of the Bayesian estimator compared to MLE and QMLE, we also see better coverage with similar higher than nominal coverage for exposure sample size nx = 5. This is similar for the σy = σx case (Fig. S11) and slightly more pronounced for the σy=15σx case (Fig. S19). Considering the performance measures, the Bayesian estimator performs better (lower values), also for the σy = σx case (Tables S2–S4) and the σy=15σx case (Table S6–S8). For the corresponding case study scenario of σy = 5σx and nx = 100, ny = 12, the Bayesian estimator clearly outperformed the MLE and QMLE. It was less biased and maintained the nominal coverage in both the two-sided and one-sided cases. For better comparison of interval lengths among the estimators, the reader is referred to Fig. S27.

Case Study

In this section, we evaluate the case study results on the basis of the simulation study results. In the case study, we have a sample size of 1,000 of exposure concentrations and a sample size of 12 of effect concentrations. We note that the exposure concentrations come from a simulation model, so that it is possible to generate an arbitrary large sample exposure concentrations. We treat the size of 1,000 exposures as being effectively of size 100.

First, we verify that the normal–normal model is not in conflict with the data. Visually, the normal distribution fits the concentration data quite well (Fig. 1). The small sample size of the effect concentrations gives low power to any formal normality tests, where the large sample size of exposure concentrations gives high power, so that even small deviations from normality are detected. Even so, we cannot reject the null hypothesis of normality (see Table S1) for either the effect or the exposure samples at a 5% significance level. In the exposure concentration data, there is some indication for non-normality evident from two of the normality tests which are only just not significant (p-values of 0.0564 and 0.0538). Even so, we take the normal–normal model as a useful model.

Table 2 Estimated risks (Rˆ), 90% two-sided confidence (credible) intervals (CI) and 95% upper confidence (credible) bounds (CB) for the MLE, QMLE (bootstrap and noncentral t), Bayesian and non-parametric estimator.

Estimator	Rˆ	90% 2-sided CI	95% upper CB	
MLE	0.0068	0.0003–0.0684	0.0684	
QMLE (noncentral t)	0.0090	0.0006–0.0784	0.0784	
QMLE (bootstrap)	0.0090	0.0002–0.0571	0.0571	
Bayes	0.0108	0.0006–0.0776	0.0806	
Empirical	0.0001	0.0001–0.0001	0.0001	

Next, we consider the estimates of the risk (Table 2). The estimates and intervals were calculated on the probit-scale and then transformed back to the original scale so as to be able to evaluate the case study results in the light of the simulation study results. For the MLE we calculated the interval estimators based on the noncentral t distribution and for the QMLE, the noncentral t and parametric bootstrap .

The non-parametric estimator was calculated using Laplace’s Law of Succession. For the sample sizes of this case study, RˆNPLLS then becomes 0+11,000⋅12+2=112,002=0.000083. We note, however, that this value is very much dependent on the sample size. A larger exposure sample size will decrease the estimate. As seen in the simulation study results, it is impossible to draw any meaningful conclusions for small risks based on the non-parametric estimator.

The three parametric estimates are similar. The bootstrap 90% confidence interval of the QMLE is clearly narrower. From the simulation study, however, the bootstrap intervals showed liberal coverage and are, therefore, less trustworthy. Considering the 95% upper confidence bound, we note that the Bayesian bound is slightly higher than that of the MLE and QMLE and higher as well than the upper bound of the Bayesian 90% credible interval. Investigating these aspects in the simulation results, we found that these differences are to be expected (see Figs. S1–S3), although the difference between the Bayesian 95% upper bound and the upper bound of its 90% credible interval is not so typical. The distances between the 95% upper bound and Rˆ as well as the ratio of the Bayesian upper bound to both the QMLE upper bound and the Bayesian upper bound of the two-sided interval fall within the respective sampling distributions as obtained in the simulations.

Based on the simulation results we, therefore, conclude that the Bayesian estimate is the most appropriate. The upper bound (0.0806) is most reliable as it has the best coverage (compared to MLE and QMLE). This is clearly seen by the black circles in the Bayesian panel in the right column of Fig. 6. This case corresponds most closely to the case study data. Based on the model and the data used, we state with 95% confidence, that the risk will not be greater than 0.0806.

Discussion and Conclusion

In this paper we studied the problem of estimating the risk for the case of small sample size for effect concentrations and small R values. The case study data showed discrepancies between the parametric and non-parametric estimators which we investigated via a simulation study. We derived and compared three parametric estimators and one non-parametric estimator for the risk. This was done under the assumption of normality for both the exposure and effect concentration data. We found that, overall, the parametric estimators have better performance than the non-parametric estimator, and the Bayesian estimator outperformed the maximum likelihood-based ones.

The Bayesian estimator in this paper was based on a non-informative prior on the underlying parameters. This resulted in a prior tendency of R toward 0.5. For small sample sizes, there was not enough data to counter this prior tendency and this resulted in an overestimation of R by the posterior mean estimator calculated on the original R scale (Figs. S4, S5, S12, S13, S20 and S21). To overcome this problem, it was needed to switch to the posterior median estimator or to switch to the probit(R) scale. We used both the probit-scale and the posterior median resulting in an estimator that outperformed its parametric counterparts. More benefit can presumably be obtained from the Bayesian estimator if we can use an informative prior, at least when the prior is not in conflict with the data. In addition, the use of probability matching priors (Datta & Sweeting, 2005) may also improve on the coverage of the credible intervals. Ventura & Racugno (2011) used a strong matching prior for Bayesian estimation of P(X > Y) based on a profile-likelihood approach.

Using the probit(R) scale in the simulation study enabled us to more easily compare the estimators for small R values. Despite giving nice statistical properties, the probit scale may not directly address a specific risk assessment question.

Comparing the parametric bootstrap and noncentral t interval estimators for the MLE and the QMLE, we found the noncentral t intervals to have better coverage. The bootstrap intervals, although a good alternative, are liberal in coverage (i.e., resulted in smaller than nominal coverage) for small sample sizes of effect concentrations. This was also found by Tian (2008).

It was clearly seen that the non-parametric estimator was not able to estimate the risk for small sample sizes and small R values. For R values above the lower bound of probit1nxny+2, the non-parametric estimator had performance measures similar to that of the MLE. As seen in our case study, however, the non-parametric estimator failed completely. The bootstrap cannot provide any variability of outcome with which to provide an interval for the estimate. In the simulation study, we also found that for small sample sizes, there was often too little variability in the data for the bootstrap to be able to quantify it (Fig. 4B). Although this translated to 0 coverage in Figs. 5 and 6, it really shows that the non-parametric estimator completely fails in these cases. For small sample sizes and small R values, therefore, we advise to use parametric estimators.

Considering the computation times of the simulation study, we note that, in addition to the Bayesian estimator being the best estimator, it can also require shorter computation time compared to the bootstrap alternatives depending on the posterior sample size. The larger posterior sample size (10,000) tends to result in slightly narrower estimates of the posterior distribution than those based on the smaller sample size (1,000). Nevertheless, the main results in Figs. 5 and 6 remain basically unchanged. In the case study, the credible interval becomes slightly wider for the larger sample size and the upper bound is slightly lower. Even so, not much is lost by taking the smaller sample size and this drastically reduces the computation time. The maximum likelihood based estimators have shorter computation time when calculating the interval estimators based on the noncentral t distribution than the bayesian estimator. The non-parametric estimator is by far the most computationally demanding due to the bootstrapping and the calculation of Eq. (4).

Assuming normality in the case and simulation study may seem as a strict assumption and going non-parametric is a way to avoid strict assumptions. For many situations in statistics the normal distribution is considered to have too light tails. In our case with very little data, going non-parametric leads to zero tails outside the range of the data. The usual area-under-the-curve-based non-parametric method can then severely underestimate the risk (often resulting in zero risk), whereas the estimate based on Laplace’s Law of Succession overestimates the risk for small true risks. To be able to draw any sensible conclusion, one has to use a parametric method. Our comparison of methods shows the advantage of using parametric methods in this case.

We conclude that making parametric assumptions, enabled us to estimate the risk for smaller sample sizes and small risks in the case the data is in fact normally distributed. Further research is needed to investigate the robustness of the parametric methods on non-normal data. We need to investigate whether semi-parametric methods and methods based on the extreme value distribution are able to estimate the tails of distributions sufficiently well from small data sets, so that they outperform the parametric methods used in this paper.

Supplemental Information

Supplemental Information 1 Supplementary Information

Click here for additional data file.

Supplemental Information 2 Exposure data (log-transformed)

Click here for additional data file.

Supplemental Information 3 Sensitivity data (log-transformed)

Click here for additional data file.

We thank Fadri Gottschalk for providing the data used in the application and Peter Craig and an anomymous reviewer for their comments that improved the manuscript.

Appendix

Result A.1 Transformation method for obtaining the pdf of a function, R = Φ(θ), from the pdf of θ.

Let θ∼Nμ,σ.

Then the pdf of R is given by fR(r) = fθ(r)|J(θ → R)|. We first obtain the Jacobian, J(θ → R): (A.1) Jθ→R=dθdR=dRdθθ=Φ−1r−1=Φ′θθ=Φ−1r−1=ϕΦ−1r−1=1ϕΦ−1r

where ϕ denotes the pdf of the standard normal distribution.

We then obtain fR(r): (A.2) fRr=fθr|Jθ→R|=fθΦ−1r1ϕΦ−1r=12πσ2exp−Φ−1r−μ22σ2112πexp−Φ−1r22=1σ2exp−Φ−1r−μ22σ2+Φ−1r22.

Result A.2 From the conditional posterior distribution of μx−μyσx2+σy2 given by μx−μyσx2+σy2∣σx2,σy2∼Nx¯−y¯σx2+σy2,σx2nx+σy2nyσx2+σy2,

we obtain the conditional posterior distribution of R (using Result A.1): (A.3) fR∣σx2,σy2r∣σx2,σy2=σx2nx+σy2nyσx2+σy2−12exp−Φ−1r−x¯−y¯σx2+σy222σx2nx+σy2nyσx2+σy2+Φ−1r22.

To obtain the marginal posterior density, fR(r), we integrate σx2 and σy2 out of the joint pdf, fr,σx2,σy2, and obtain the required result. (A.4) fRr=∫0∞∫0∞fr,σx2,σy2dσx2dσy2=∫0∞∫0∞fr∣σx2,σy2fσx2fσy2dσx2dσy2=∫0∞∫0∞σx2nx+σy2nyσx2+σy2−12exp−Φ−1r−x¯−y¯σx2+σy222σx2nx+σy2nyσx2+σy2+Φ−1r22× bxaxΓaxσx2−ax−1exp−bxσx2byayΓayσy2−ay−1exp−byσy2dσx2dσy2

with ax=nx−12

ay=ny−12

bx=nx−1sx22

by=ny−1sy22.

Additional Information and Declarations

Competing Interests

Author Contributions

Cajo J.F. ter Braak is an Academic Editor for PeerJ.

Rianne Jacobs analyzed the data, contributed reagents/materials/analysis tools, wrote the paper, prepared figures and/or tables, reviewed drafts of the paper.

Andriëtte A. Bekker and Hilko van der Voet contributed reagents/materials/analysis tools, wrote the paper, reviewed drafts of the paper.

Cajo J.F. ter Braak analyzed the data, contributed reagents/materials/analysis tools, wrote the paper, reviewed drafts of the paper.

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
