# Peer review of "Parametric estimation of P(X > Y) for normal distributions in the context of probabilistic environmental risk assessment"

_PeerJ, doi:10.7717/peerj.1164_

## Round 0.1 · original submission · Minor Revisions

· Academic Editor

Minor Revisions

Two reviewers agree that this paper is well written and a contribution to a more robust probabilistic risk assessment. It should be quite a straightforward revision. However I would ask the authors specifically to address the comment if the approach is a "sledgehammer" maybe for a simple issue.

Reviewer 1 ·

Basic reporting

-

Experimental design

-

Validity of the findings

-

Additional comments

This is a very well written paper that should be published. The statistical computations seem from a short viewing – and with some reasonable effort of a review – performed well. This interesting contribution, however at some points perhaps uses a bit a statistical sledgehammer to crack a nut. Below are some general suggestions relating to this and other contexts.

Of course the few effect concentrations used are not normally distributed (contrary to the assumption made in the manuscript). It's a bit difficult to assert that they are, in such a pronounced way and with such great computational effort/scenarios. Why not just indicate, perhaps more cautiously, that for a parametric modelling exercise on normal distributions, the raw effect concentration data is simply fed into normal distributions? In this context, we must also see that the “under the curve” interpretation of risk does not cover everything as clearly as the authors suggest at the beginning, at least, as soon as the curves are not normally shaped. Verdonck et al. (2003), Environmental Toxicology and Chemistry, Vol. 22, comprehensively show the critical points and limitations of under the curve approaches. In addition, the fundamental assumption, per se, on “normally” distributed data, based on a few data points, also requires that such an assumptive basis for the model is even more clearly underlined in the comparison/discussion of the performance of the different estimators. I would give more weight to this and would underline it more clearly in the discussion section.

I would also consider the estimators as options, with pros and cons, and perhaps not focus too much on making rankings. From a pragmatic point of view, all four estimators show as very low and, in principle, identical risk levels (mostly slightly below 0.01), and they all also fully correlate to a nonparametric reference study from which the data is taken. Table 1 does not refer correctly to this reference study; among other things, I found that the other study reports, in Table 2, a silver risk of 0.007 in waters and not a zero risk, as the authors suggest. This changes the comparison to the other work, which is itself very cautious in interpreting risk results. The statement on pg. 14, where we are 95% sure that the risk will not be higher than 8%, is exactly as “imprudent” (a word used by the authors themselves in such contexts) as saying that the risk is zero if the model shows no overlap of effects and exposure. I would always state that the model shows something, but not that we are sure, for example, “based on the model and the data used, we may state that…” In this context of risk results interpretation, I also found interesting statements in the other reference study, on its last page: To assess the acceptability of risk, …short-term or geographically limited PEC extremes or small groups of highly sensitive species...

I would also integrate the central results of this paper, shown in Figure 5 and Figure 6, into one single figure for a better overview and comparison of those outcomes.

·

Basic reporting

Line 66: missing number in section reference.

Line 73: suggest insert "when parameter values are known or assumed" at the end of the sentence.

Line 81 and more occurrences later: "AUC under the ROC curve" is not quite right since AUC="Area under the curve". I would suggest just writing "area under the ROC curve

Line 103-104: citing Aldenberg (2002) and Verdonck et al (2003) for this result is very "ecotox-centric". This could be said to be standard statistical theory (second year undergraduate maths in the UK). It would certainly be preferable to cite Reiser and Guttman (1986) which you cite elsewhere.

Line 137 and 138: "Inverse-gamma" is currently not formatted as text. Suggest using \mbox (or \text) in LaTeX

Line 171: The distinction between confidence interval and credible interval is not just a matter of language. Putting credible in parentheses is I think misleading. Better to write "credible intervals for Bayesian methods and confidence intervals for others".

Section 2.3: You should explain that all the confidence interval procedures you use do not claim to deliver the correct coverage: they are all explicitly approximate. Similarly, you should explain that Bayesian credible intervals do not claim a coverage frequency although some Bayesian procedures have very good coverage properties.

Line 220 and un-numbered preceding lines: it is not clear which scenario was used for the pilot simulation.

The R code provided in the supplementary material is exemplary for its clarity and formatting. To be completely transparent, it would be good to include the overall R script which obtained all the samples and then produced the figures.

There is a considerable literature in Bayesian statistics on "frequency-matching priors" or "probability matching priors". See the review by Datta and Sweeting (2005) and more recent literature. Since your evaluation is based substantially on coverage, it would be natural to choose a frequency-matching prior were one to be available. At the very least, you should cite the literature and state that it might be possible to improve the coverage of your Bayesian method by using a different prior. In particular, Ventura and Racugno (2011) consider P[X<Y] and claim a "strong-matching" prior. However, their results are for assuming sigma_x=sigma_y although they claim that it would be easy to generalise.

Experimental design

Line 146: rather than "posterior sample size of 1000", I would write "Monte Carlo sample size of 1000".

But more seriously: are the histograms in figure 2 really produced from a Monte Carlo sample size of 1000. Intuitively (and having done a small experiment), I think the sample size must have been larger to produce such a close match of histograms to theoretical pdfs. Moreover, 1000 is not a large Monte Carlo sample for estimating either a 90% HPD interval or the 95th percentile which you need later. A small sampling experiment shows that for a standard normal, estimates of the 95th percentile based on samples of 1000 values range from 1.53 to 1.75 (90% of cases). That seems quite a large error in the context of your work. I see no reason why you could not use a substantially larger number of Monte Carlo samples (and I think you did so for figure 2).

Validity of the findings

In section 3.1, you draw a conclusion that the agreement on probit scale between median, mode and mean for the Bayesian method is a reason to use probit scale to evaluate the performance of all methods. I find this reasoning unconvincing. What probit scale does here is make differences between small values of R look important since it stretches out small values of R. From the point of view of scientific understanding of a particular application, I think that it is genuinely interesting to know whether R is 0.01 or 0.001 or 0.0001. From a risk assessment point of view, this is less clear: once R is small enough, it is not particularly interesting to know exactly how small it is. I am not suggesting that you fundamentally change the paper: from a mathematical perspective, there are strong reasons for looking at something like probit scale when comparing estimator performance. But I do think that you should be clear that the probit scale may not directly address the risk assessment question and that it is therefore possible that an estimator which performs less well on probit scale might actually turn out be be better for risk assessment in practice. Ideally, you would augment the existing analysis with a comparison based on some form of decision valuation such as a loss function which measures how bad the consequence is of mis-estimating true R by estimate r.

Para at line 409: you describe the bootstrap intervals as "liberal". I did not understand what you intended. To me, the word liberal would suggest that the bootstrap intervals tend to be wider than needed but my reading for figures 5 and 6 is that the bootstrap coverage is less than it should be, especially for small sample sizes. I suggest that you find a different wording.

Para at line 410: The paragraph is critical of the non-parametric estimator but I think that it should be more so. Coverage and interval length in figures 5 and 6 are a a good overall comparison of methods but they do not reveal that a method sometimes fails altogether. Your case study shows that the non-parametric method can fail disastrously: the bootstrap cannot provide any variability of outcome with which to provide an interval for the estimate. So even when R is above the lower bound given, the samples will sometimes be such that no interval is obtained; this will happen less often for higher R but it is still a big weakness.

Additional comments

Line 52: "In the risk assessment literature ...". I suggest that you insert "ecotoxicological" before "risk".

I found it quite hard in figures 5 and 6 to make comparisons between different estimators with respect to the vertical scale. I do not have a recommendation to make for an improvement but I thought that you should be aware that there is an issue.

---

## Round 0.2 · accepted · Accept

· Academic Editor

Accept

The paper has been revised accordingly, and can now be published.